# The Collaborative IPD of Sleep and Stillbirth (Cribss): is maternal going-to-sleep position a risk factor for late stillbirth and does maternal sleep position interact with fetal vulnerability? An individual participant data meta-analysis study protocol

Minglan Li,[1] John M D Thompson,[1,2] Robin S Cronin,[1] Adrienne Gordon,[3,4] Camille Raynes-Greenow,[5] Alexander E P Heazell,[6,7] Tomasina Stacey,[8] Vicki Culling,[9] Victoria Bowring,[10] Edwin A Mitchell,[2] Lesley M E McCowan,[1] Lisa Askie[11]

For numbered affiliations see end of article.

**Correspondence to**
Dr Minglan Li;
m.li@auckland.ac.nz

## ABSTRACT

**Introduction** Accumulating evidence has shown an association between maternal supine going-to-sleep position and stillbirth in late pregnancy. Advising women not to go-to-sleep on their back can potentially reduce late stillbirth rate by 9%. However, the association between maternal right-sided going-to-sleep position and stillbirth is inconsistent across studies. Furthermore, individual studies are underpowered to investigate interactions between maternal going-to-sleep position and fetal vulnerability, which is potentially important for producing clear and tailored public health messages on safe going-to-sleep position. We will use individual participant data (IPD) from existing studies to assess whether right-side and supine going-to-sleep positions are independent risk factors for late stillbirth and to test the interaction between going-to-sleep position and fetal vulnerability.

**Methods and analysis** An IPD meta-analysis approach will be used using the Cochrane Collaboration-endorsed methodology. We will identify case–control and prospective cohort studies and randomised trials which collected maternal going-to-sleep position data and pregnancy outcome data that included stillbirth. The primary outcome is stillbirth. A one stage procedure meta-analysis, stratified by study with adjustment of a priori confounders will be carried out.

**Ethics and dissemination** The IPD meta-analysis has obtained central ethics approval from the New Zealand Health and Disability Ethics Committee, ref: NTX/06/05/054/AM06. Individual studies should also have ethical approval from relevant local ethics committees. Interpretation of the results will be discussed with consumer representatives. Results of the study will be published in peer-reviewed journals and presented at international conferences.

**PROSPERO registration number** CRD42017047703.

### Strengths and limitations of this study

► Late stillbirth is a rare event in high-income countries, and individual participant data meta-analysis of several studies can yield a sufficiently large sample size for exploring interactions and subgroup analysis that are difficult to undertake within a single study.

► There is no restriction on language or countries where the study was conducted, therefore the results from this study are likely to be generalisable.

► Consumer representatives will oversee the conduct of the study. Their involvement will help to design appropriate research questions and will help the implementation and translation of the research outcomes.

► One limitation of the study is that the maternal going-to-sleep positions are likely to be self-reported.

## INTRODUCTION

Stillbirth, the death of a baby before birth, is a major global burden affecting more than 2.6 million families per year.[1] In high-income countries, the rate of late stillbirth (28 weeks or greater) varies widely from 1.3 to 8.8 per 1000 births[2] and is approximately twice as common as neonatal death.[3] Furthermore, the annual rate of reduction for neonatal death is twice that of stillbirth.[2] The variations between countries suggest it is possible to further reduce late stillbirth. Importantly, maternal characteristics present in early pregnancy only explain a small amount of the risk for late stillbirth.[4] Therefore, significant

reductions in late stillbirth require identification of additional maternal risk factors amenable to modification during pregnancy.[5]

Accumulating evidence suggests that supine going-to-sleep position may be a modifiable risk factor for stillbirth in late pregnancy. Stacey *et al* first reported an association between going-to-sleep position and late stillbirth, with women who did not go-to-sleep on their left side, the night before the baby was suspected to have died, having an increased odds of stillbirth.[6] Among non-left-sided sleepers, the odds were greater in women who went to sleep supine; and there was also a borderline increase in odds in women who went to sleep on their right side.[6] Similar associations between supine going-to-sleep position and late stillbirth have since been reported by several studies.[7–9] In addition to the epidemiological evidence, a number of physiological studies have suggested that the relationship between supine going-to-sleep position and late stillbirth is biologically plausible. Significant haemodynamic changes in maternal and fetal circulation have been observed in relation to maternal position in late pregnancy, with decreased maternal cardiac output and uterine blood flow,[10] and pulsatility index in the fetal middle cerebral artery (a surrogate for fetal hypoxia)[11] seen in maternal supine position when compared with left position. A recent study by Stone *et al* has shown that when the mother is in the supine position, the fetus spends more time in behavioural state 1 (fetal quiescence) and less time in active fetal behavioural state 4, compared with when the mother is on her left side, indicating supine position may be a mild hypoxic stressor.[12] It was hypothesised that these physiological changes associated with supine position are related to the direct compression of the inferior vena cava by the gravid uterus.[13] Furthermore, supine sleep position is also associated with sleep disturbed breathing and obstructive sleep apnoea,[14] which have also been associated with pregnancy complications such as pre-eclampsia, fetal growth restriction[15] and gestational diabetes.[15 16] These pregnancy complications are known risk factors for stillbirth[17] and might represent another mechanism that contributes to the association between supine going-to-sleep position and late stillbirth.

The findings from the epidemiological studies combined with the supportive physiological evidence suggest that the association between supine sleep position and late stillbirth is likely to be causal. Informing pregnant women and their healthcare providers about optimal going-to-sleep position in late pregnancy is a strategy that may reduce stillbirth and is potentially harmless. Therefore, there is an urgent need to assess the accumulated evidence to develop a public health campaign. However, there are some unanswered questions that are critical for developing clear public health messages. First, it is unclear whether right-sided going-to-sleep position is a risk factor for late stillbirth. A borderline increase in risk was reported with right side compared with left side going-to-sleep position in the Stacey *et al*[6] study. However, this association was not found in other studies.[7 9] The inconsistent finding of right side going-to-sleep position warrants further clarification so that clear advice about whether women should be advised to go-to-sleep on either side or only on their left side can be developed. Second, there is no evidence whether there are groups of women who are at further increased risk when they go-to-sleep in a suboptimal position (such as those who smoke, are overweight or have small babies, etc.) and how other stillbirth risk factors interact with sleep position. Stillbirth is the end point of diverse pathological processes. Multiple risk factors and pathological events can contribute at different time points and cumulatively lead to the final event. Our research group has hypothesised a triple-risk framework for late stillbirth that cannot be explained by one risk factor or condition alone.[18] We speculate that three groups of factors namely maternal factors (eg, obesity, smoking), fetal and placental factors (eg, a small for gestational age (SGA) fetus) and an additional stressor(s) (eg, reduced uterine blood flow associated with supine position) in themselves may be insufficient to cause the death, but their combination may have a lethal effect.[18] Individual stillbirth case–control studies published to date have insufficient power to explore fully the interactions between supine going-to-sleep position, markers of fetal vulnerability and adverse maternal factors. Furthermore, it is important to explore other factors that may also be associated with supine sleep position such as SGA, reduced fetal movements and sleep disturbed breathing, as this may provide insight into the potential mechanism of risk associated with the supine position.

The Collaborative IPD Sleep and Stillbirth (Cribss) group was established in December 2016. We aim to synthesise the current evidence about going-to-sleep position and stillbirth risk. Additionally we will address the above unanswered questions by combining and analysing the individual participant data (IPD) from all available studies in an IPD meta-analysis. IPD meta-analysis is considered the gold standard approach to evidence synthesis as it has the potential to improve the precision and reliability of the results obtained from individual studies.[19] In contrast to the traditional approach of meta-analysis, which extracts summary (aggregate) data from study publications, an IPD meta-analysis uses line-by-line original data sourced directly from the researchers responsible for the relevant studies. An IPD meta-analysis involves the central collection, checking, harmonisation and reanalysis of the original data of all eligible participants from each of the available studies. With proper quality assessment and standardisation processes, an IPD meta-analysis can model complex relationships, which traditional meta-analyses are not able to do.[20] It is particularly useful in evaluating multi-factorial frameworks by evaluating critical outcome determinants and their interactions.

## OBJECTIVES

The main questions to be addressed by the Cribss IPD meta-analysis are:

1. Is maternal going-to-sleep position associated with late stillbirth?
2. Are indicators of fetal vulnerability, including: maternal obesity, SGA, maternal smoking, maternal second-hand tobacco exposure, substance use, alcohol consumption, maternal medical conditions (including pre-existing hypertension and diabetes), and maternal perception of fetal movements associated with late stillbirth, and does maternal going-to-sleep position interact with indicators of fetal vulnerability to influence the risk of late stillbirth? Does birth weight centile interact with maternal going-to-sleep position to influence the risk of late stillbirth?

Secondary questions to be addressed by the first cycle of Cribss IPD meta-analysis are:

1. Is sleep disturbed breathing associated with late stillbirth? Is (are) going-to-sleep position(s) associated with greater risk of late stillbirth in women with sleep disturbed breathing? Is sleep disturbed breathing a moderator for sleep position in relation to late stillbirth?
2. Are factors that may influence vena caval compression (eg, long sleep duration, sleeping during the day, restless legs) associated with risk of late stillbirth? Do these factors interact with going-to-sleep position?
3. Do women who report they received advice about sleep position have lower risk of late stillbirth compared with women who did not receive such advice?
4. Do women who report they received advice about awareness of fetal movements have a lower risk of late stillbirth than women who did not receive such advice?

## METHODS AND ANALYSIS

This study will apply an IPD meta-analysis approach, and will follow the methodology endorsed by the Cochrane Collaboration where applicable.[21 22] We will adhere to the Preferred Reporting Items for Systematic Reviews and Meta-analyses IPD statement for reporting findings. The study will be conducted by Cribss group which comprises the participating study investigators, an IPD expert and consumer representatives. The coordination centre is located in the Department of Obstetrics and Gynaecology at the University of Auckland, Auckland, New Zealand.

### Eligibility criteria

Study inclusion criteria (regardless of whether the study is published or unpublished):

► Case–control and prospective cohort studies which collected:
  – maternal going-to-sleep position during pregnancy
  – pregnancy outcome data that included stillbirth and

  – aimed to recruit controls with an ongoing pregnancy at similar gestation to the cases.
► Randomised controlled trials which collected:
  – maternal going-to-sleep position during pregnancy
  – pregnancy outcome data that included stillbirth.

Participant-level exclusion criteria:

► Multiple pregnancy in the third trimester.
► Major congenital abnormality at study entry or major congenital abnormality as a cause of death found post study entry or postrandomisation in randomised controlled trials.
► Gestation less than 28 weeks when last sleep position data during pregnancy was collected.
► Termination of pregnancy at greater than or equal to 28 weeks.
► Received study intervention that might have an impact on going-to-sleep position.

### Information sources and search strategy

We will develop the search strategy according to the Cochrane Collaboration guidelines prior to the initial literature search. A search of the databases: MEDLINE, EMBASE, LILACS, Web of Science, OpenGrey and Google Scholar will be conducted for the purpose of locating published research about an association between maternal sleep position and late pregnancy stillbirth. We will also access the WHO International Clinical Trials Registry Platform to identify any ongoing and registered trials. Proceedings from International Stillbirth Alliance annual conferences and The International Society for the Study and Prevention of Perinatal and Infant Death international conferences will be manually searched. Published perinatal conference abstracts will also be identified through the above database searches. Experts in the field and the collaborative group will be asked about their knowledge of any unpublished studies. To increase the likelihood of identifying all relevant studies the reference lists of all retrieved articles will be hand searched. No language restriction will be applied.

Four search terms will be used to search the databases with the article title, abstracts and body all searched. The search terms are: 'stillbirth', 'fetal death', 'perinatal death' and 'sleep' and synonyms. The search terms will be tested to check that they effectively located the types of articles that are consistent with the inclusion criteria prior to conducting the search in all engines. An example of a detailed MEDLINE search strategy is presented in online supplementary appendix 1.

### Selection process

Study eligibility will be assessed independently by two members of the Cribss group, any disagreements will be adjudicated by a third member. Eligibility assessment will be based on published protocols, method sections from publications, and unpublished protocols and, or study information requested from potential eligible study investigators. All potential eligible study investigators will be contacted to verify eligibility. Participant level exclusion

criteria will be applied during the analysis. The main investigator and/or the corresponding author from any eligible study will be approached via email to participate in the Cribss IPD meta-analysis study. If there is no reply, other coauthors of the published manuscript will be subsequently approached.

### Data acquisition and data management

The data centre is located in the Department of Obstetrics and Gynaecology at the University of Auckland, New Zealand, who will manage transferring and sharing of data. A detailed data management plan has been reviewed and agreed by all Cribss members.

Each eligible study lead investigator will be asked to provide deidentified individual level participant data for each participant enrolled in their study. Some indirect potential identifiers (eg, age, ethnicity) are essential demographic characteristics, and will be required. A study ID for each participant will be retained as this is essential for data integrity checking and data cleaning. Each study investigator will also be asked to provide metadata (such as questionnaires, data collection forms, data dictionaries) and study-level data to explain the variables, and data on the study representativeness (box 1).

The anonymised data in a common format (eg, .cvs, .xls or other formats that can be converted by the Cribss data centre) will be requested for transfer via the University of Auckland institutional Seafile file synchronisation and share platform or equivalent secure means. The Seafile platform has built-in file encryption. Files are encrypted before syncing to the server. User authentication is needed to access the files.[23]

The anonymised dataset from each participating study will be checked for data integrity. This will include: (1) checking data range and outliers, (2) clarifying missing data, (3) identifying invalid values, (4) detecting duplicates and (5) verifying internal consistency where appropriate. Reports of discrepancies will be generated and sent to each participating study investigator for further verification or correction where necessary.

After appropriate data cleaning, the individual participating study investigators will confirm and sign-off on their own dataset before it is merged into the IPD database. New variables will be generated following a set of consistent harmonisation rules that will be decided by the Cribss group. An IPD data dictionary will be created to document the details of variables (including variable names, type, explanation and validation rules) to help other users to understand the dataset.

### Data items

We aim to collect the data items from each participating study (see box 1).

### Outcome measures

The primary outcome is late stillbirth, using the WHO recommended definition for stillbirth for international comparison: 'a baby born with no signs of life at or after

---

**Box 1    Data items will be requested from participating studies**

**Study-level information**
1. Study inclusion and exclusion criteria.
2. Matching method of cases and controls.
3. Time period of recruitment.
4. Number of cases and controls.
5. Informed consent procedure.
6. Study participant representativeness (eg, minimal demographic data comparison between participant and eligible non-participant, or between participants and a relevant comparison of a maternity care population).

**Participant-level information**
A. Maternal characteristics
  1. Unique study ID.
  2. Maternal demographic details including: age, ethnicity.
  3. Obstetric history.
  4. Maternal height.
  5. Earliest available maternal weight in the study pregnancy.
  6. Gestation at earliest available weight.
  7. Last available maternal weight in current pregnancy.
  8. Gestation at last available weight.
  9. Study centre (if the study was conducted in more than one centre).
  10. Highest completed education level at the time of recruitment.
  11. Marital status at the time of recruitment.
  12. Pre-existing medical conditions and medical conditions during the study pregnancy.
  13. Smoking status before and during the study pregnancy.
  14. Exposure to second-hand smoke before and during the study pregnancy.
  15. Alcohol consumption before and during the study pregnancy.
  16. Recreational drug usage before and during the study pregnancy.
B. Maternal sleep practices and fetal movement data in every available time frame
  1. Going-to-sleep position.
  2. Sleep duration.
  3. Number of times getting up during the night (eg, to go to the toilet).
  4. Frequency of daytime napping.
  5. Bed size.
  6. Number of people shared bed with.
  7. Self-reported details of snoring behaviour.
  8. Insomnia.
  9. Sleep quality as measured by validated questionnaire.
  10. Maternal perception of fetal movement.
  11. Advice received on fetal movement.
  12. Advice received on sleep position.
C. Antenatal care and pregnancy outcomes
  1. Gestation (gestation at enrolment for controls, and gestation at diagnosis of stillbirth for cases).
  2. Baby sex.
  3. Baby birth weight.
  4. Gestation for calculating birthweight centile.
  5. Birthweight centile per original study standards.
  6. Type of facility of baby's birth.
  7. Gestation at earliest ultrasound.
  8. Blood pressure and gestation at measurement.
  9. Type of maternity provider.
  10. Number of antenatal visits in each trimester.

## Box 1 Continued

11. Ultrasound scans (first trimester scan, anatomy scan and third tri-mester growth scan(s)).
12. Antenatal vaginal bleeding.
13. Hospital admission(s).
14. Use of antibiotics.
15. Nutritional supplements.
16. Clinical suspicion of fetal growth restriction (FGR)/small for gesta-tional age (SGA).
17. Management of clinically suspected FGR/SGA.
18. Laboratory tests for glucose metabolism (including Polycose glu-cose challenge test, haemoglobin A1c and oral glucose tolerance test), hepatitis B status and blood group and the gestation that the tests were conducted.

D. Stillbirth cases specific data
1. Time of day mother thought the baby died.
2. The reason that the mother thought something was wrong with the pregnancy.
3. The reason that the mother saw a health practitioner at the diagno-sis of stillbirth.
4. Maternal decision on postmortem.
5. Placental pathology results.
6. The Perinatal Society of Australia and New Zealand coding for clas-sification of cause of stillbirth.

28 weeks' gestation'.[24] Intrapartum stillbirth will be included in the analysis with the rationale that supine going-to-sleep position may result in a vulnerable baby that is unable to tolerate labour.

### Risk of bias assessment

Risk of bias for non-randomised studies will be assessed in duplicate and independently by two investigators from the Cribss group, using Risk of Bias In Non-randomised Studies–of Exposure assessment tool.[25] The assessment results will be compared. Any disagreement will be resolved by discussion or by a third reviewer.

### Statistical analysis plan

A detailed statistical analysis plan for the main questions has been prepared by the Cribss data centre group and reviewed, and agreed on by the Cribss group prior to the analysis (online supplementary appendix 2). All going-to-sleep positions will be compared with left-sided going-to-sleep position as the reference group. The last available going-to-sleep position during pregnancy (within 2 weeks before stillbirth in cases) will be harmonised and used for the primary objectives.

An IPD analysis will be performed. A one stage approach to analysis will be taken so that the IPD from all eligible studies are included in a single model. Logistic regression models will be used for the binary outcome (late stillbirth). A fixed study effect and a study site effect will be included in the model specification as strata. Univariable analysis will be performed to evaluate the association between sleep position and late stillbirth risk. The interaction between sleep position and factors indicating a vulnerable preg-nancy will be assessed in bi-variable models. A multivariable model will be developed incorporating previously reported confounders and any significant interaction terms, once it has been established what confounders can be controlled for consistently across studies. Estimate of risk will be reported as OR and 95% CIs. We will also explore if sleep apnoea is a moderator for sleep position in relation to late stillbirth using moderator analyses.

If an important confounder is not available for one or more studies, sensitivity analysis will be conducted, with and without these studies, to compare risk estimates. If there are any controls who reported their pregnancy going-to-sleep position after they have given birth, sensi-tivity analysis will be conducted without these controls. Where sufficient data exist, all analysis will be also conducted in term and preterm subgroups. For missing data in each individual study, no imputation will be carried out. Statistical analyses will be performed using SAS V.9.3 (SAS Institute).

## ETHICS AND DISSEMINATION

The participating studies retain the right to withdraw their data from the analysis at any time. Final IPD results will be presented to the nominated representative from each participating study prior to publication and public dissemination. Interpretation of the results will be discussed with the Cribss consumer representatives. Results of the study will be published in peer-reviewed journals and presented at national and international conferences. For the publications from the main ques-tions, every Cribss member will participate in the manu-script preparation and editing. Authorship will be guided by the recommendations of the International Committee of Medical Journal Editors.

## CONCLUSION

Cribss is the first IPD meta-analysis to evaluate the current evidence of the relationship between maternal going-to-sleep position and late stillbirth. The study will allow assessment of important interactions that cannot be tested in standard, aggregate data meta-analysis. The overall goal of Cribss is to reduce late stillbirth by devel-oping high-quality data-based evidence to inform public health messages about optimal late pregnancy sleep prac-tices. This IPD meta-analysis may identify subgroups of women at greater risk (such as those with known SGA fetuses, who continue to smoke during pregnancy or are overweight) and thus develop evidence that can be used to tailor public health messages.

**Author affiliations**
[1]Department of Obstetrics and Gynaecology, University of Auckland, Auckland, New Zealand
[2]Department of Paediatrics and Child Health, University of Auckland, Auckland, New Zealand
[3]Department of Newborn Care, Royal Prince Alfred Hospital Women and Babies, Sydney, New South Wales, Australia
[4]Charles Perkins Centre, The University of Sydney, Sydney, New South Wales, Australia

[5]Sydney School of Public Health, The University of Sydney, Sydney, New South Wales, Australia

[6]Division of Developmental Biomedicine, Faculty of Medical and Human Sciences, Maternal and Fetal Health Research Centre, University of Manchester, Manchester, UK

[7]St. Mary's Hospital, Central Manchester University Hospitals NHS Foundation Trust, Manchester Academic Health Science Centre, Manchester, UK

[8]School of Healthcare, University of Leeds, Leeds, UK

[9]Vicki Culling Associates, Auckland, New Zealand

[10]Stillbirth Foundation, Annandale, New South Wales, Australia

[11]National Health and Medical Research Council Clinical Trials Centre, University of Sydney, Camperdown, New South Wales, Australia

**Contributors** ML, JMDT, RSC, AG, CR-G, AEPH, TS, EAM, LMEM and LA conceptualised the study. ML, JMDT, RSC, AG, CR-G, AEPH, TS, VC, VB, EAM, LMEM and LA have participated in study design and funding application. ML drafted the manuscript and appendix 2. RSC drafted appendix 1. LA, JMDT, RSC, AG, CR-G, AEPH, TS, EAM and LMEM critically revised the manuscript. ML, JMDT, RSC, AG, CR-G, AEPH, TS, VC, VB, EAM, LMEM and LA have read and approved submission of the final manuscript. LMEM is the guarantor of the review.

**Funding** This work was supported by 2016 Trans-Tasman Research Funding Grant by Cure Kids and Red Nose, Australia (Grant 6601).

**Disclaimer** Funder has no role in developing the protocol.

**Competing interests** None declared.

**Patient consent** Not required.

**Ethics approval** New Zealand Health and Disability Ethics Committee, ref: NTX/06/05/054/AM06.

**Provenance and peer review** Not commissioned; externally peer reviewed.

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
