## [Reviewer comments · BMJ Open]

ARTICLE DETAILS

TITLE (PROVISIONAL)	The Collaborative IPD of Sleep and Stillbirth (Cribss): Is maternal going-to-sleep position a risk factor for late stillbirth and does maternal sleep position interact with fetal vulnerability? – An Individual Participant Data Meta-Analysis study protocol
AUTHORS	Li, Minglan; Thomprn, John; Cronin, Robin; Gordon, Adrienne; Raynes-Greenow, Camille; Heazell, Alexander; Stacey, Tomasina; Culling, Vicki; Bowring, Victoria; Mitchell, Edwin; McCowan, Lesley; Askie, Lisa

VERSION 1 – REVIEW

REVIEWER	Ingela Rådestad Professor at Sophiahemmet University, Sweden
REVIEW RETURNED	06-Nov-2017

GENERAL COMMENTS	This will require a lot of work and I appreciate that you will do. My expectations for new knowledge primarily concerns the secondary end points, and I hope you have the strength to deal with them as thoroughly as you will do with the primary endpoints.
---

REVIEWER	Sotirios Saravelos 1. The Chinese University of Hong Kong, Hong Kong 2. London Deanery, UK
REVIEW RETURNED	24-Nov-2017

GENERAL COMMENTS	Very comprehensive outline of the proposed IPD meta-analysis including all the required information: background, scope, methodology, ethics, data collection/protection and analysis. It is certainly an important topic of which the authors have a proven track record in addressing in the recent years. As it will undoubtedly draw international scientific and media attention, I wonder whether the authors have considered a strategy for limiting the dissemination of sensational headlines that may invariably appear following such work.
---

REVIEWER	Mark Simmonds Centre for Reviews and Dissemination University of York UK
REVIEW RETURNED	08-Jan-2018

GENERAL COMMENTS	This protocol is well written, and is methodologically sound, using all methods appropriate for a systematic review and meta-analysis of IPD. I have only a few minor comments and suggestions for clarification. (Objective 2) Do you genuinely intend to look at the possible impact of all the named factors on stillbirth? You cannot possibly have all the relevant studies or data for such analysis, because many studies of these factors will not have reported sleep position. This objective needs to be clarified. If you do intend to investigate these factors it must be discussed in the statistical analysis section (which currently only mentions sleep position), and how you will avoid bias from excluding relevant studies must be considered. I assume you intend to identify all RCTs that reported sleep position and stillbirth, regardless of intervention (so not just studies randomising women to sleep position). If so, will you use only the control-arm data? If you intend to use intervention-arm data how will you account for the intervention effect in your analyses? Will you include any case-control study, or just matched C-C studies? In any analysis of observational data proper accounting for potential confounders is critical. While I agree that you cannot completely decide on all confounding variables at this stage, I think you must give some more details to avoid bias due to post-hoc selection. You should ideally specify all possible confounders that would be adjusted for (provided all studies report them), and those confounders that it is essential to adjust for (so studies would be excluded if not reporting them). This could be set out in Table 1. Ideally you should compare all sleep positions to each other, and not just to the baseline of left-side sleeping. You state that the impact of sleep position might be because it increases the risk of sleep apnoea, GDM etc. Do you intend any “moderator” analyses to investigate this? That is, to investigate if sleep-position increases the risk of sleep apnoea and hence stillbirth, and whether the impact of sleep position is only due to its impact on sleep apnoea etc., or whether it has other modes of effect.
---

VERSION 1 – AUTHOR RESPONSE

Thank you editors and reviewers.

Editors comments:

- Please revise the title of your manuscript to include the research question, study design and setting. This is the preferred format of the journal.

RE: The title has now been revised as: The Collaborative IPD of Sleep and Stillbirth (Cribss): Is maternal going-to-sleep position a risk factor for late stillbirth and does maternal sleep position interact with fetal vulnerability? - An Individual Participant Data Meta-Analysis study protocol.

- Please revise the ‘Strengths and limitations’ section of your manuscript. This section should relate specifically to the methods, and should not include a general summary of the study.

RE: ‘Strengths and limitations’ section has now been revised as suggested. (Lines 69-72 has been taken out).

Reviewer(s)' Comments to Author:

Reviewer: 1

Reviewer Name: Ingela Rådestad

Institution and Country: Professor at Sophiahemmet University, Sweden

Please state any competing interests or state 'None declared': None declared

Please leave your comments for the authors below

This will require a lot of work and I appreciate that you will do. My expectations for new knowledge primarily concerns the secondary end points, and I hope you have the strength to deal with them as thoroughly as you will do with the primary endpoints.

RE: Thank you for the comments. We will ensure the same rigor is used for analysis of secondary endpoints. As stated in the manuscript, all datasets from each participating study will be checked for data integrity on five aspects, regardless of whether they are primary or secondary endpoints.

Participating study investigators will receive reports regarding their original data distribution and collection.

Reviewer: 2

Reviewer Name: Sotirios Saravelos

Institution and Country: 1. The Chinese University of Hong Kong, Hong Kong 2. London Deanery, UK

Please state any competing interests or state 'None declared': None

Please leave your comments for the authors below

Very comprehensive outline of the proposed IPD meta-analysis including all the required information: background, scope, methodology, ethics, data collection/protection and analysis.

It is certainly an important topic of which the authors have a proven track record in addressing in the recent years.

As it will undoubtedly draw international scientific and media attention, I wonder whether the authors have considered a strategy for limiting the dissemination of sensational headlines that may invariably appear following such work.

RE: Thank you for the comment. As stated in the manuscript (Line 337), interpretation of the results will be discussed with the Cribss consumer representatives. The Cribss group will draft a responsible media release that will be in line with the authors' interpretation of results including strength and limitations of the study.

Reviewer: 3

Reviewer Name: Mark Simmonds

Institution and Country: Centre for Reviews and Dissemination, University of York, UK

Please state any competing interests or state 'None declared': I have previously collaborated with one author (L Askie) on an IPD review. No other interests to declare.

Please leave your comments for the authors below

This protocol is well written, and is methodologically sound, using all methods appropriate for a systematic review and meta-analysis of IPD. I have only a few minor comments and suggestions for clarification.

(Objective 2) Do you genuinely intend to look at the possible impact of all the named factors on stillbirth? You cannot possibly have all the relevant studies or data for such analysis, because many studies of these factors will not have reported sleep position. This objective needs to be clarified. If you do intend to investigate these factors it must be discussed in the statistical analysis section (which currently only mentions sleep position), and how you will avoid bias from excluding relevant studies must be considered.

RE: Thank you for the comment. We aim to collect all named factors as possible confounders, and aim to explore the potential interactions between sleep position and these factors. It is not our objective to assess these risk factors separately. Question 2 and 3 have now been merged into one question.

I assume you intend to identify all RCTs that reported sleep position and stillbirth, regardless of intervention (so not just studies randomising women to sleep position). If so, will you use only the control-arm data? If you intend to use intervention-arm data how will you account for the intervention effect in your analyses?

RE: Thank you for the suggestion. We will not use intervention arm data from RCTs that include an intervention likely to modify the exposure. The eligibility criteria section has now been revised accordingly (Line 207-218).

Will you include any case-control study, or just matched C-C studies?

RE: We will include any case control studies that aimed to recruit controls with an on-going pregnancy. We will use unconditional logistic regression to analyse the combined data.

In any analysis of observational data proper accounting for potential confounders is critical. While I agree that you cannot completely decide on all confounding variables at this stage, I think you must give some more details to avoid bias due to post-hoc selection. You should ideally specify all possible confounders that would be adjusted for (provided all studies report them), and those confounders that it is essential to adjust for (so studies would be excluded if not reporting them). This could be set out in Table 1.

RE: Thank you for the comment. We have drafted a detailed statistical plan including all planned definitions for confounding variables. This document has now been attached as an appendix (revised in line 303-305).

Ideally you should compare all sleep positions to each other, and not just to the baseline of left-side sleeping.

RE: Comparing to left-side sleep position is our primary hypothesis, and is in line with the physiological plausibility. The attached analysis plan has included "Depending on the similarity of the risk estimates, going-to-sleep position may be further merged to fewer groups such as supine vs non-supine groups in the analysis of interaction".

You state that the impact of sleep position might be because it increases the risk of sleep apnoea, GDM etc. Do you intend any "moderator" analyses to investigate this? That is, to investigate if sleep-position increases the risk of sleep apnoea and hence stillbirth, and whether the impact of sleep position is only due to its impact on sleep apnoea etc., or whether it has other modes of effect.

RE: Thank you for the comment. It is a good research question. We will explore measure of sleep apnoea in moderator analyses if data are available from participating studies. (revised in line 178-179 and 320-321)

VERSION 2 – REVIEW

REVIEWER	Mark Simmonds Centre for Reviews and Dissemination, University of York, UK
REVIEW RETURNED	12-Feb-2018
GENERAL COMMENTS	I am happy with the changes made to the protocol in response to my previous review, and have no further comments.